# Screening and Demulsification Mechanism of Fluorinated Demulsifier Based on Molecular Dynamics Simulation

**DOI:** 10.3390/molecules27061799

**Published:** 2022-03-09

**Authors:** Xiaoheng Geng, Changjun Li, Lin Zhang, Haiying Guo, Changqing Shan, Xinlei Jia, Lixin Wei, Yinghui Cai, Lixia Han

**Affiliations:** 1College of Petroleum Engineering, Southwest Petroleum University, Sichuan 610500, China; lichangjunemail@sina.com; 2College of Chemical Engineering and Safety, Binzhou University, Binzhou 256600, China; guohaiying1987@126.com (H.G.); sdscq@163.com (C.S.); 18434362466@126.com (X.J.); 3School of Petroleum Engineering, Northeast Petroleum University, Daqing 163318, China; zl980110@163.com (L.Z.); weilixin73@163.com (L.W.); 4Chambroad Chemical Industry Research Institute Co., Ltd., Binzhou 256505, China; yinghui.cai@chambroad.com (Y.C.); lixia.han@chambroad.com (L.H.)

**Keywords:** demulsifier, fluorinated, demulsification mechanism, molecular dynamics simulation, neural network analysis, genetic function approximation

## Abstract

In order to solve the problem of demulsification difficulties in Liaohe Oilfield, 24 kinds of demulsifiers were screened by using the interface generation energy (IFE) module in the molecular dynamics simulation software Materials Studio to determine the ability of demulsifier molecules to reduce the total energy of the oil–water interface after entering the oil–water interface. Neural network analysis (NNA) and genetic function approximation (GFA) were used as technical means to predict the demulsification effect of the Liaohe crude oil demulsifier. The simulation results show that the SDJ9927 demulsifier with ethylene oxide (EO) and propylene oxide (PO) values of 21 (EO) and 44 (PO) reduced the total energy and interfacial tension of the oil–water interface to the greatest extent, and the interfacial formation energy reached −640.48 Kcal/mol. NNA predicted that the water removal amount of the SDJ9927 demulsifier was 7.21 mL, with an overall error of less than 1.83. GFA predicted that the water removal amount of the SDJ9927 demulsifier was 7.41mL, with an overall error of less than 0.9. The predicted results are consistent with the experimental screening results. SDJ9927 had the highest water removal rate and the best demulsification effect. NNA and GFA had high correlation coefficients, and their R^2^s were 0.802 and 0.861, respectively. The higher R^2^ was, the more accurate the prediction accuracy was. Finally, the demulsification mechanism of the interfacial film breaking due to the collision of fluorinated polyether demulsifiers was studied. It was found that the carbon–fluorine chain had high surface activity and high stability, which could protect the carbon–carbon bond in the demulsifier molecules to ensure that there was no re-emulsion due to the stirring external force.

## 1. Introduction

Liaohe Oilfield in China mainly produces heavy oil and super heavy oil. It is very difficult to demulsify the oil emulsion because of its large asphaltene content and high viscosity [1]. Liaohe Oilfield has entered the middle and late stage of exploitation. Since water flooding cannot meet the demand of the oilfield production increase, alkali–surfactant–polymer (ASP) flooding technology can better meet the demand of the production increase in Liaohe Oilfield in the middle and later stages of production [2,3]. ASP flooding technology can greatly improve oil recovery, but the synergistic effect and emulsifying effect in the process of oil displacement make the emulsification of produced fluid very serious [4,5]. When the crude oil is produced by ASP flooding, a large number of surfactants, polymers and other chemicals are used, resulting in more complex produced fluid systems and more difficult demulsification [6]. Transportation and refining of this stable emulsion without treatment can cause serious problems, such as pipeline corrosion, scaling, increased equipment load and fuel consumption [7,8]. Conventional demulsification technology mainly includes physical demulsification, chemical demulsification and biological demulsification [9,10,11]. Chemical demulsification requires simpler equipment and has a lower cost and better demulsification effect. It can be used alone or combined with other demulsification methods to achieve efficient demulsification [12,13]. However, at present, many demulsifiers cannot meet the actual needs of Liaohe Oilfield or are difficult to apply due to cost, safety and other factors [14]. Therefore, the synthesis of a kind of demulsifier which is suitable for the development of Liaohe Oilfield and can efficiently and rapidly demulsify has become an urgent problem, which is of great significance to the good and efficient development of Liaohe Oilfield.

In recent years, molecular dynamics simulation technology has developed rapidly and is gradually applied to surfactants such as demulsifiers. Molecular dynamics simulation refers to the use of computer technology, discusses the interfacial structure and interfacial action of emulsion after adding the demulsifier at the molecular level in order to explain the role of demulsifiers through this technology, optimizes the selection of efficient demulsifiers and better serves oilfield production [15,16]. Using molecular dynamics simulation to guide experimental research not only makes the experimental data and their universal mechanism more visible, but also provides a new direction for future experimental research.

Marquez et al. [17] first studied the demulsification behavior of demulsifiers at the oil–water interfacial film of oil-in-water emulsion using an atomic model. They found that surfactants that can be used as demulsifiers must have the following characteristics: Firstly, the solubility of demulsifiers in the aqueous phase must be higher than that in the oil phase. Secondly, they must have certain diffusions and concentrations. Finally, the surface activity of demulsifiers must be higher than that of emulsifiers. The demulsifier with the above characteristics can reach the oil–water interface film and reduce the stability of the interface film to achieve demulsification.

Ballal et al. [18] used the improved iSAFT (interfacial statistical association fluid theory) to explore the influence of poly (ethylene oxide)–propylene oxide polyether on the interfacial film of water–toluene by studying the molecular weight, the ratio of EO to PO, branching degree and order degree, so as to understand the influence of demulsifier structure on the interfacial film at the molecular level and predict the performance of real demulsifiers. The results show that the interfacial tension decreased with the increase in molecular weight and the number of branched chains. When EO:PO = 1:1, the interfacial tension is at its minimum. Moreover, the surface activity of PEO-PPO-PEO is higher than that of PPO-PEO-PPO.

Zhang et al. [19] used a polyamide-amine dendrimer demulsifier to study the effect of the hydrophobic chain on interfacial properties and demulsification with molecular dynamics simulation technology. The results show that with the increase in the demulsifier concentration, the kinetic parameters n and t* obtained by characterizing the molecular diffusion rate decreased. At the same time, unlike the traditional demulsifier adsorption and diffusion behavior, with the increase in the hydrophobic chain length, the t* value decreased and the n value increased, showing a slow diffusion–adsorption process.

A machine model algorithm can predict and integrate new rules and development trends from a large number of data texts in multiple dimensions. In general, the process of using machine algorithm to simulate experimental data can be divided into two steps: inputting old data and simulating new trends. With the development and widespread application of computer algorithms, researchers often use the neural network algorithm and genetic algorithm to predict the mixed-phase pressure, and good prediction results have been achieved.

The purpose of this study was to provide efficient and economical fluorinated polyether demulsifiers for Liaohe Oilfield. Compared with general demulsifiers, the fluorine atoms contained in this demulsifier can partially or completely replace the hydrogen atoms on the hydrocarbon chain, so that the nonpolar groups in the demulsifier can form carbon–fluorine bonds with stronger bond energy, and this carbon–fluorine chain with higher bond energy can show strong stability. Fluorinated polyether demulsifiers have better surface activity, chemical stability, thermal stability and compatibility than conventional demulsifiers. Fluorinated hydrocarbon groups are also hydrophobic, which can reduce pollution. The interfacial generation energy (IFE) in molecular dynamics was used to screen 24 kinds of demulsifiers. Neural network analysis (NNA) and genetic function approximation (GFA)were applied to predict demulsification, so as to look for the rules from the existing experimental data to obtain the corresponding prediction conclusions.

## 2. Experimental

### 2.1. Materials

Tetraethylenepentaamine was purchased from Beijing Tianyu Kanghong Chemical Technology Co., Ltd. (Beijing, China). P-trifluoromethyl phenol was purchased from Shanghai Sahn Chemical Technology Co., Ltd. (Shanghai, China). Formaldehyde was purchased from Shanghai Macklin Biochemical Technology Co., Ltd. (Shanghai, China). Xylene and toluene were ordered from Shanghai Jizhi Biochemical Technology Co., Ltd. (Shanghai, China). Potassium hydroxide was purchased from Shanghai Sibaiquan Chemical Co., Ltd. (Shanghai, China). Potassium hydroxide was purchased from Shanghai Sibaiquan Chemical Co., Ltd. Ethylene oxide (EO) and propylene oxide (PO) were purchased from Zibo Shandong Zixiang Sales Chemical Co., Ltd. (Zibo, China). The tested oil sample was produced from fluid from a block in Liaohe Oilfield. Fluorinated demulsifiers were synthesized by using trifluoromethyl phenol, formaldehyde and other raw materials as initiators and then synthesized through polymerization reaction with propylene oxide and ethylene oxide [20]. The physiochemical characteristics are shown in Table 1.

### 2.2. Molecular Optimization and Model Construction

All the simulations were performed on the molecular dynamics software Materials Studio2018. The interaction parameters of surfactants came from the condensed-state optimized molecular force field—COMPASS force field.

Firstly, the 3D model structures of n-decane and demulsifier molecules were built by using the Visualizer module in the program, and the geometric optimization of the structures of the three surfactant molecules was carried out by using the Smart method through the COMPASS force field of the Dmol3 module, so that the surface molecular system could achieve the minimum energy, and the optimized molecular structure of the optimal molecular conformation was obtained, as shown in Figure 1.

Then, the crude oil system model and demulsifier system model were established at 278 K by using the construction tool under AC module, COMPASS force field and Periodic Cell periodic boundary conditions. Based on the position reference of the rectangular coordinate system, the size of the system box was set. With the origin as the center, the lengths in x, y and z directions were 4 nm × 4 nm × 12 nm, respectively. The system model is shown in Figure 2. The simulation systems with different EO/PO ratios were composed of 2000 n-sunane molecules and 500 water molecules.

Finally, the Dynamics tool under the Forcite module was used. The simulation level was MEDIUM, and the simulation system ensemble was NVT ensemble, keeping the system at a constant temperature of 298 K. AtomBased was used to represent van der Waals interaction and electrostatic interaction. Andersen method was selected for parameter control of ensemble, namely temperature control. In addition, Berendsen method was used for pressure change. A 3000-step process was established and the last nanosecond result was obtained by statistical analysis.

### 2.3. Neural Network Analysis (NNA)

Neural network analysis (NNA) refers to the method of machine learning and data processing that controls various parameters and layers based on the way in which neurons in the brain of organisms transmit information [21,22]. NNA was first proposed by McCulloch and Pitts in 1943 as a way to simulate the analysis of neurons in the brain. Although it makes too many assumptions and simplifications than real brain neurons, it still contributes considerable intelligence in research. Therefore, NNA has considerable research and application value. Since then, NNA has been greatly developed, and hundreds of models have been proposed. Figure 3 is a complete typical three-layer neural network structure, which is divided into three parts: multinode input layer, single-node output layer and hidden layer. A three-layer BP neural network can solve almost all the prediction problems near exact precision, so only one hidden layer was used in this study.

Neurons refer to nodes, which are the most basic structure of neural networks. Each node is an information-processing element. In addition to the input layer, each node uses the transformed linear combination of node output from the lower layer as its input:(1)$$I_{i}=\sum_{j}w_{ij}X_{j}+\theta_{i}$$

In the expression, *I_i_* is the input to the *i* node, *X_j_* is the output of the *j* node in the previous layer, *j* is the summation of all nodes in the previous layer, *w_ij_* is the connection weight between nodes, and *θ_i_* is the offset value. It is worth noting that the nodes between each layer are not fixed, which needs to be set according to the actual situation. When exploring MMP, the nodes in the input layer are designed as multiple variables that affect MMP, and the output is the corresponding MMP value. The number of hidden layers and the number of nodes in each layer can be defined by users themselves, so as to make the output results closer to the real value.

The transfer function needs to be realized through the transfer function between the input layer and the hidden layer and between the hidden layer and the output layer. In addition, the conversion information is realized by setting the weights and bias values. In this way, the data between the input layer and the output layer can be connected and their relationship can be directly reflected. The study uses a transfer function called Sigmoid transfer function (Formula (2)), which allows for easy differentiation and has a smooth function to achieve data output in a narrow range.
(2)$$y=\frac{1.2}{1+e^{-x}}-0.1$$

After setting the above information, training should be started and the neural network should be learned independently. The training minimizes the error and makes the final prediction more accurate. Here, BFGS algorithm is used to find the minimum value of the error, and the error function (3) is used to determine the matching degree between the calculated output and the expected output:(3)$$E=\sum_{i=1}^{n}C_{i}{\left(y_{i}-y_{i}^{\prime }\right)}^{2}+Q\sum_{j}{\left(x_{j}-\bar{x_{j}}\right)}^{2}+P\sum_{k,l}w_{k,l}^{2}$$
where *C_i_* is the parameter value of the proportion of results. In this study, the item is 1, $y_{i}$ and $y_{i}^{’}$ represent the true value and the predicted value, respectively. *Q* is the penalty factor set for the missing value, $x_{j}$ is the missing data value of the system guess, $\bar{x_{j}}$ is the average value of each input data, *P* represents the penalty factor of connection weight, and $w_{k,l}$ represents the connection weight. The first item is the main item of the error, which is the sum of squares of the difference between the predicted value and the actual output value of the model. The second item represents the error caused by filling the missing data. In addition, the average connection weight is added to the error function to prevent the collapse or error caused by excessive weight. In this way, the learning cycle iteration of the neural network can be carried out until the error drops to a certain level, and finally a trained neural network can be obtained.

### 2.4. Genetic Function Approximation (GFA)

Genetic function approximation (GFA) is carried out through selection, crossover and mutation [23,24]. This algorithm is based on Darwin’s theory of evolution and some viewpoints of genetics. Generally speaking, choosing an excellent father will lead to better offspring. In addition, the mutation operation accelerates the progress of the algorithm and does not fall into local optimum [25]. By applying this algorithm to the field of intelligence, the optimal selection method is obtained to improve the economic benefits. The structural process is shown in Figure 4.

The parallel method can improve the convergence speed under the premise of ensuring a certain fitness, so as to obtain more accurate results. The method extended by this method is the genetic function approximation method. This method has good performance, including high robustness, and can be used to improve fault tolerance.Therefore, this method has been widely used in practical applications.

The core idea of GFA is similar to that of GA (genetic algorithm), which encodes the region to be searched as one or more strings, each string representing a position in the search space, where each group of strings is called a population, and the evolution of the population makes it move towards the search target. At the beginning of the setting, the initial clock group is set to 100, and through 500 iterations, in the iterative process, there is a choice of crossover and mutation. The members through variation need to score; namely, the following fitness function is used to score (Formula (4)). In GFA, the evaluation criteria of the model are related to the quality of data regression fitting. The fitness function used in this study was Friedman LOF function:(4)$$F=\frac{SSE}{M{\left[1-\lambda (\frac{C+dp}{M})\right]}^{2}}$$
where *SSE* is the sum of squares of errors, *C* is the number of items in the model, which is not a constant, p is the total number of descriptors contained in all model items, *M* is the number of samples in the training set, *λ* is a safety factor, the value is 0.99, to ensure that the denominator of the expression does not become zero, and *d* is the scaling smoothing parameter; the following expression is associated with the specified scaling LOF smoothing parameter:(5)$$d=\alpha (\frac{M-C_{\max}}{C_{\max}})$$

*C*_max_ means the maximum equation length.

Selection is not random; individuals with good adaptability are chosen. After the reproduction of the selected individuals, the new members need to be graded to determine whether they are the next selected object. Crossing process is the exchange of genetic information between parent chromosomes. The selection of genetic information in the cross process is random. When the cross selection is over, the new members need to be graded to determine whether they are remixed into the population to seek better results.

Parents:(6)$$x_{1}^{2},x_{2}|x_{4},x_{3}^{2}$$
(7)$$x_{1},x_{3}|x_{4},x_{5}^{2}$$

Child:(8)$$x_{1}^{2},x_{2}|x_{4},x_{5}^{2}$$

In order to make genetics more scientific, mutation operation is needed. What is reflected in the computer is the mutual transformation between 0 and 1 so as to find the optimal solution faster.

Finally, through a series of genetic operations, the offspring are optimized, and the mutation operation is used to prevent the calculation results from falling into local optimum, leading to wrong results. The optimal population obtained by this method was the optimal solution we found

### 2.5. Turbidity Point and HLB Value Test

The cloud point of fluorinated polyether demulsifier was determined by Cintra 10e UV-Vis spectrometer (GBC Scientific Instruments Company, Melbourne, Australia). The HLB (hydrophilic–lipophilic balance) value of demulsifier was calculated according to the cloud point of surfactant and the empirical formula of HLB to obtain the corresponding HLB value. The empirical formula is as follows:*HLB* = 0.0980X + 4.02(9)

X is the cloud point value of 1wt% fluorinated polyether demulsifier.

### 2.6. Experiment on Demulsification and Water Removal of Demulsifier

Crude oil emulsion with water content of 17.02% was placed into constant-temperature water bath heated to 55 °C for 30 min and then put into a stirring motor for 8 min at a speed of 2000 r/min. After that, it was put into the stirring machine for 5 min. A total of 50 mL crude oil emulsion was poured into a calibrated test tube and put it into a water bath heated to 60 °C and kept at a constant temperature for 25 min. Care was taken to ensure the height of the water surface did not exceed the height of the crude oil in the test tube. The demulsifier was added into the test tube with micropipeter, and the cork was tightened. The test tube was turned upside down and shaken 3–5 times, and the cork was loosened to let off air. The bottle was recorked, and the tube was shaken 150 times by hand to fully mix the demulsifier and crude oil emulsion. After the cork was capped, the bottle was placed in a water bath at 60 °C for settling. The volume of dehydration at different times was observed to obtain the dehydration volume V. The blank sample without demulsifier addition was set to obtain the dehydration amount V_b_. Therefore, the dehydration amount after adding demulsifier was V_d_ = V−V_b_.

Demulsification efficiency is calculated as follows:


Efficiency (%) = (V_O_ − V_d_)/V_O_ × 100

where V_O_ is the volume of water (water content) in the crude oil emulsion and V_d_ is the volume of water remaining in the oil phase after demulsifier addition.

## 3. Results and Discussion

### 3.1. Molecular Dynamics Simulation Results

The energy optimization trend and optimization steps of molecular dynamics simulation of some demulsifiers are shown in the Figure 5, Figure 6, Figure 7, Figure 8, Figure 9, Figure 10, Figure 11, Figure 12, Figure 13, Figure 14, Figure 15, Figure 16, Figure 17 and Figure 18. 

When the system was balanced, the free energy of the solution reached its minimum. Therefore, there was a chemical potential equilibrium relationship between surfactant monomer and micelle:(10)$$\mu_{g}^{0}+KT\ln X_{g}=g(\mu_{l}^{0}+KT\ln X_{l})$$
where $\mu_{l}^{0}$ is the standard chemical potential of surfactant monomer, $X_{l}$ is the molar composition of the surfactant monomer, $\mu_{g}^{0}$ is the standard chemical potential of micelles containing g surfactant monomers, and $X_{g}$ is the molar composition of surfactant micelles.

Surfactants are dispersed in water in a molecular state, and their hydrophobic ends are arranged at the water interface to form a glacier structure, which reduces the entropy of the system [26]. However, when the hydrophobic end of the surfactant leaves the water interface, the surfactant molecules associate and the glacier structure is destroyed; then, the water molecules are separated from the bondage, thereby increasing the entropy of the system [27]. The formation process of micelles is a spontaneous entropy-driven process. In this process, the chaos of the system increases, and the total formation energy becomes negative [28].

The interfacial generation energy (IFE) refers to the reduced energy of the system after the surfactant molecules enter the oil–water interfacial layer. The stability of the interface can be investigated, and its value is closely related to the interaction force between surfactant and water molecules, surfactant molecules, surfactant and oil molecules. The calculation formula is as follows:(11)$$IFE=\frac{E_{total}-(n\times E+E_{\mathrm{ref}})}{n}$$
where $E_{total}$ is the total energy of the surfactant system after equilibrium at the oil–water interface, Kcal/mol. $E_{\mathrm{ref}}$ is the energy of oil–water interface system without the demulsifier, Kcal/mol (−42933.07). *N* is the number of demulsifier molecules in the system. E is the potential energy of the demulsifier molecule, Kcal/mol.

Table 2 shows that IFE value was negative, indicating that the energy of the whole system decreased. Therefore, after analyzing the absolute value of IFE, it was determined 6# demulsifier had the best effect.

### 3.2. NNA and GFA Prediction Results

The prediction results of 24 demulsifiers are shown in Table 3. NNA and GFA were used to predict the demulsification effect of 24 demulsifiers, and the results are shown in Table 4. As shown in Table 3, where X and Y are PO and EO values, demulsification dehydration represents a demulsification effect.

Firstly, demulsification experiments were carried out for 24 kinds of demulsifiers to obtain the actual demulsification effect, and the results are shown in Table 3. Then, NNA and GFA were used to predict the demulsification effect of 24 demulsifiers, respectively, and the results are shown in Table 4. As shown in Table 3, X and Y are the values of PO and EO, and the amount of water removal represents the demulsification effect. Figure 19 is the molecular structure of fluorinated demulsifier.

The calculation results for the HLB value are shown in Table 3.

The values of x and y in R_3_ are given in Table 4.

It can be seen from Figure 20 and Figure 21 that GFA had a slightly higher correlation coefficient, but from Table 5, it can be found that the correlation coefficients of both were lower than 0.9, and both were above 0.8. In general, these two methods can predict the demulsification effect of this type of demulsifier. The square value of the correlation coefficient (R^2^, coefficient of determination) reflects that the greater the R^2^, the stronger the predictive ability.

The GFA prediction formula can be edited into:Water removal amount = 0.626504940 × RAMP (65.555188173 − X) + 0.069567811 × RAMP(78.604247624 − Y) − 0.012461417 × [RAMP (74.580034129-X)]^2^ − 0.001476625 × [RAMP (70.127414082 − Y)]^2^ + 5.110992651(12)

X and Y are the number of EOs and POs in the experiment, RAMP is the slope function.

With this prediction function, such molecules can be predicted through the formula, and the values of different proportions of X and Y and their dehydration rates can be roughly known, which can greatly save time.

### 3.3. Demulsification Mechanism

Substances that ensure oil–water phase dispersion and do not interfere with each other are called oil–water interfacial films [27]. The formation mechanism of an oil–water interface film is mainly as follows: Natural emulsifiers such as asphaltene and colloid in crude oil emulsion are stably adsorbed on the surface of water droplets, forming an interfacial film with low surface tension and interfacial free energy [28]. The demulsification mechanism of fluorinated polyether demulsifier in this study was the main mechanism of breaking the interface film. With the large-scale use of polymer demulsifiers, the mechanism of breaking interfacial film is increasingly recognized by a large number of researchers [29]. This kind of polymer surfactant has been favored by many oil fields because of its economy. When it is applied to specific crude oil demulsification, the dosage is very small, and the demulsification is very high [30].

The fluorinated polyether demulsifier developed in this paper is a nonpolar surfactant, which introduces a fluorine atom instead of a hydrogen atom to a hydrocarbon chain. The bond energy of the C-F bond is higher than that of the C-H bond, but the polarity is lower than that of the C-H bond [31,32]. Due to the characteristics of the fluorocarbon chain, compared with ordinary demulsifiers, it can reduce the oil–water interfacial tension more rapidly, accelerate the aggregation of water droplets and has better demulsification effect. The demulsification mechanism is essentially that surfactant molecules replace and break the interfacial film to release captured oil particles. Surfactants are added to the emulsion, and because of its higher interfacial activity, they replace the natural emulsifier molecules, such as asphaltene and colloid, adsorbed on the oil–water interfacial film and rearrange the oil–water interface, resulting in the rapid coalescence of water droplets and the realization of oil–water separation. The hydrophilicity of the hydrophilic block (PEO) of the block polyether demulsifier is higher than that of asphaltene molecules in oil. Therefore, the hydrophilic block (PEO) of the block polyether demulsifier can rapidly replace asphaltene molecules at the oil–water interface. When subjected to heating or shaking, the Brownian motion of macromolecules in the emulsion is intensified, and the number of collisions between macromolecules is increased. Therefore, the unstable interfacial film formed by demulsifier molecules is broken. Demulsifier molecules have higher stability because of the high bond energy of the C-F bond and the shielding property of the C-C, which ensures that there is no re-emulsion due to excessive stirring and other factors. Figure 22 depicts a diagram of the demulsification mechanism of the demulsifier.

## 4. Conclusions

In this paper, based on the physical parameters of Liaohe crude oil emulsion, 24 kinds of demulsifiers were screened by using the interface generation energy (IFE) module in the molecular dynamics simulation software Materials Studio, and neural network analysis (NNA) and genetic function approximation (GFA) were applied to predict demulsification. The simulation results show that the SDJ9927 demulsifier 6# had the largest reduction in the total energy of the oil–water interface and the strongest reduction in oil–water interfacial tension, and the interfacial formation energy reached −640.48 Kcal/mol. NNA predicted that the water removal amount of the SDJ9927 demulsifier was 7.21 mL, with an overall error of less than 1.83. GFA predicted that the water removal amount of the SDJ9927 demulsifier was 7.41 mL, with an overall error of less than 0.9. The predicted results are consistent with the experimental screening results. SDJ9927 had the highest water removal amount and the best demulsification effect. NNA and GFA had high correlation coefficients, and their R^2^s were 0.802 and 0.861, respectively. The higher R^2^ was, the more accurate the prediction accuracy was. Finally, the demulsification mechanism of the fluorinated polyether demulsifier was the following: The demulsifier molecules with high interfacial activity replace the natural emulsifier on the oil–water interfacial film and form a new unstable interfacial film. When subjected to heating or shaking, the interfacial film collides with other macromolecules, and the interfacial film breaks, and the water droplets gather to complete the oil–water separation. The demulsification mechanism of the interfacial film was broken by the collision of the fluorinated polyether demulsifier. It was found that when subjected to heating or shaking, the macromolecules in the emulsion exhibited irregular Brownian motion and collided with other macromolecules, resulting in the rupture of the interfacial film. The water in the internal phase broke through the interfacial film and entered the external phase to aggregate, so as to achieve the purpose of oil–water separation.

## Figures and Tables

**Figure 1 molecules-27-01799-f001:**
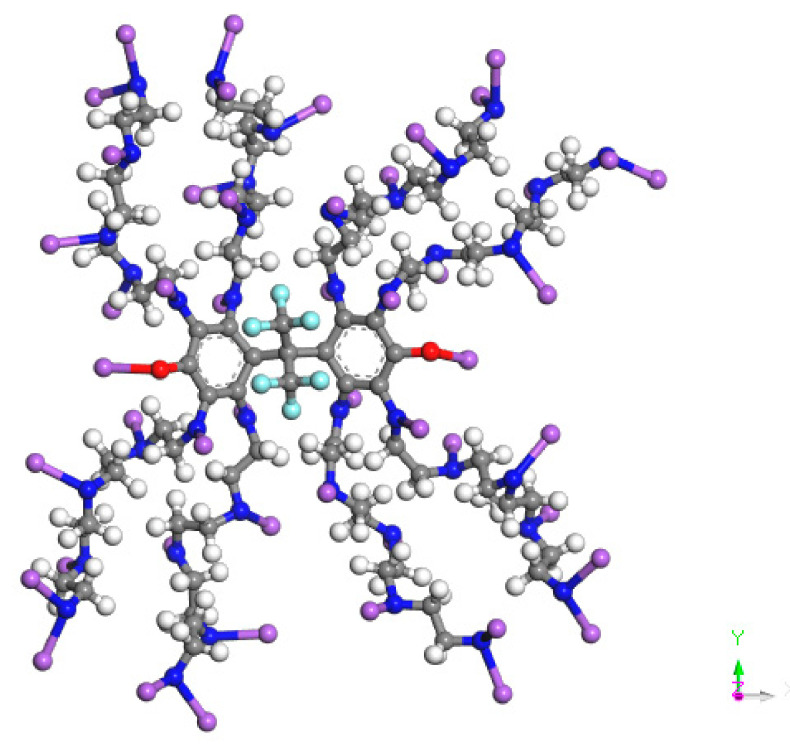
Schematic diagram of optimized molecular structure, where blue is N, red is O, white is H, gray is C, and purple is R_3_ = (C_3_H_6_O)_x_(C_2_H_4_O)_y_.

**Figure 2 molecules-27-01799-f002:**
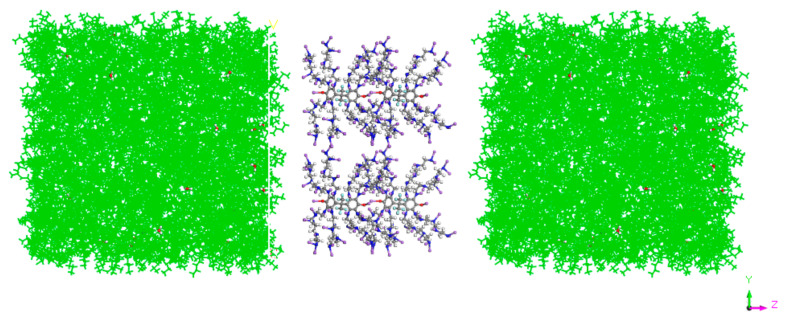
Demulsifier system model.

**Figure 3 molecules-27-01799-f003:**
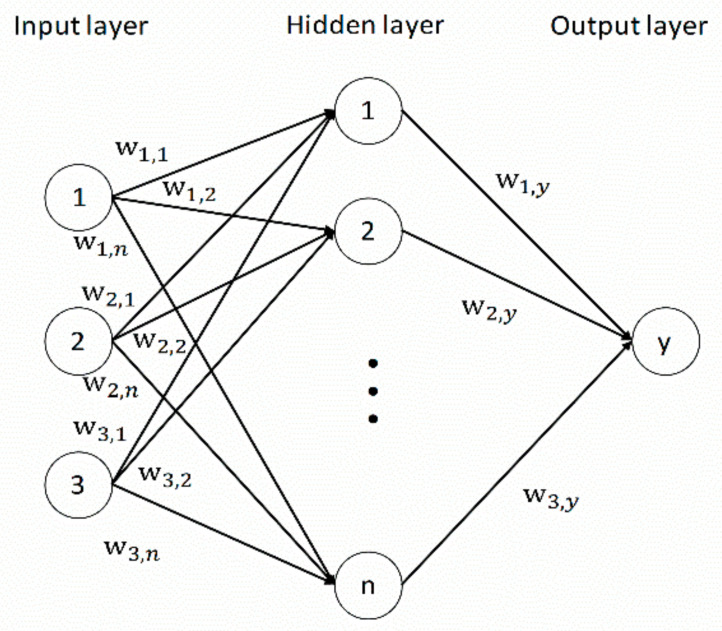
Basic structure of three-layer neural network.

**Figure 4 molecules-27-01799-f004:**
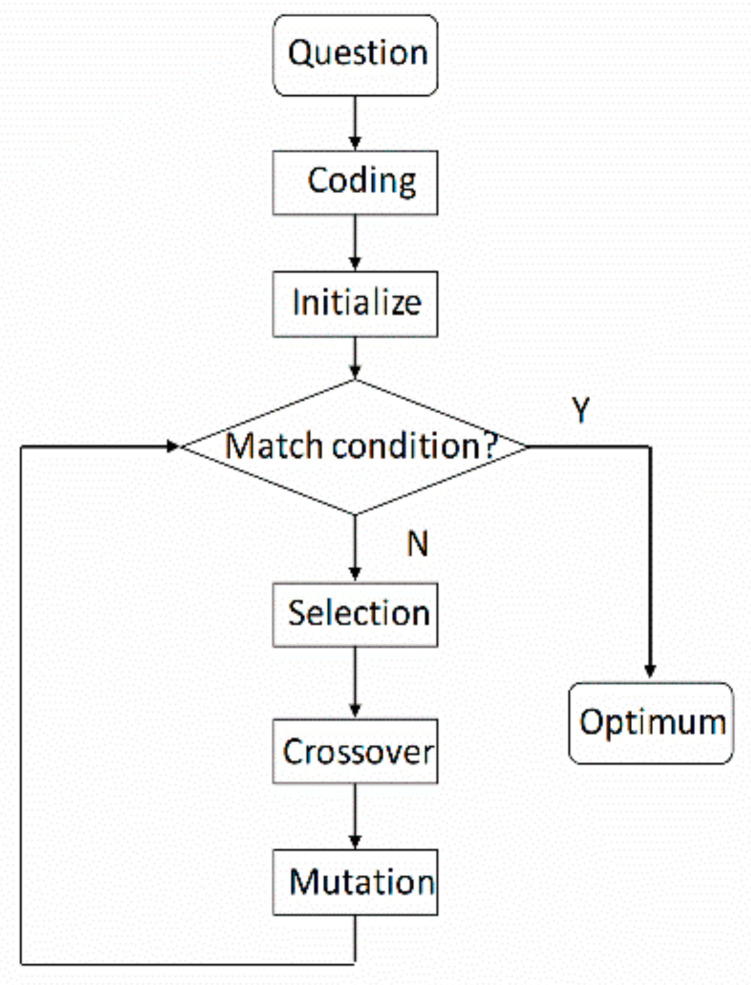
Genetic function approximation chart.

**Figure 5 molecules-27-01799-f005:**
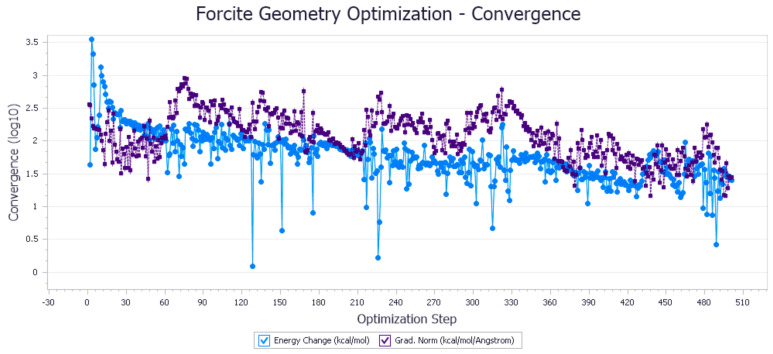
Demulsifier2 # oil–water interface geometric optimization steps.

**Figure 6 molecules-27-01799-f006:**
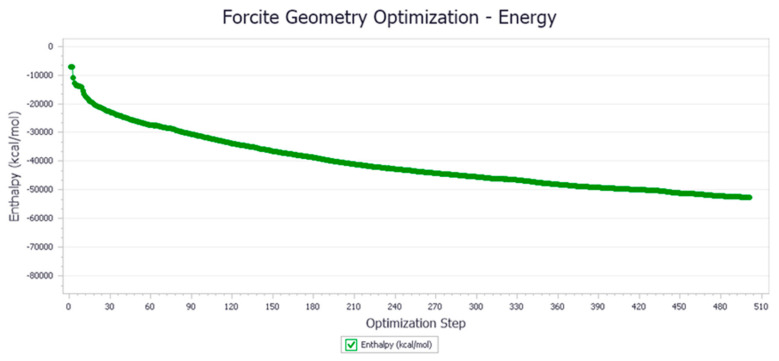
Demulsifier2 # oil–water interface geometry optimization energy trend.

**Figure 7 molecules-27-01799-f007:**
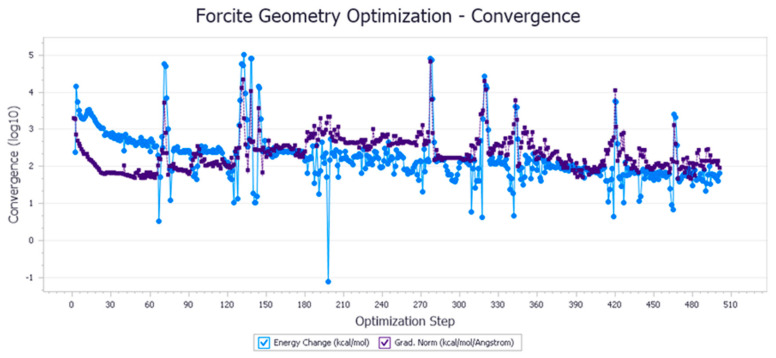
Demulsifier4 # oil–water interface geometric optimization steps.

**Figure 8 molecules-27-01799-f008:**
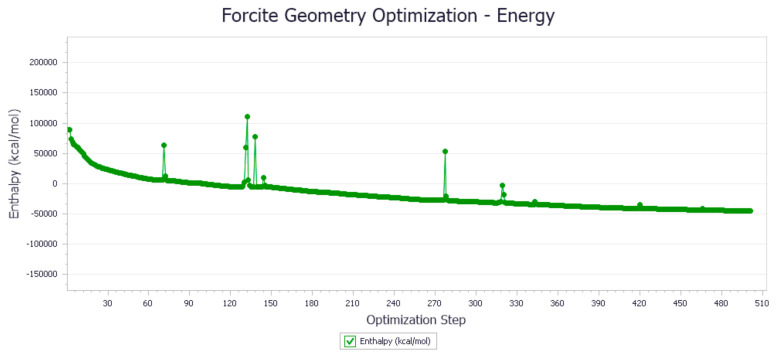
Demulsifier4 # oil–water interface geometry optimization energy trend.

**Figure 9 molecules-27-01799-f009:**
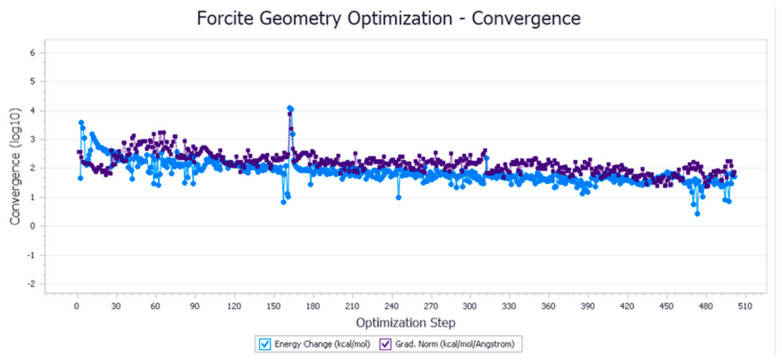
Demulsifier6 # oil–water interface geometric optimization steps.

**Figure 10 molecules-27-01799-f010:**
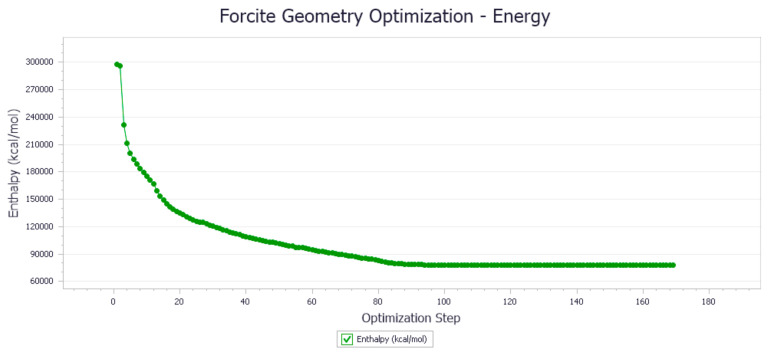
Demulsifier6 # oil–water interface geometry optimization energy trend.

**Figure 11 molecules-27-01799-f011:**
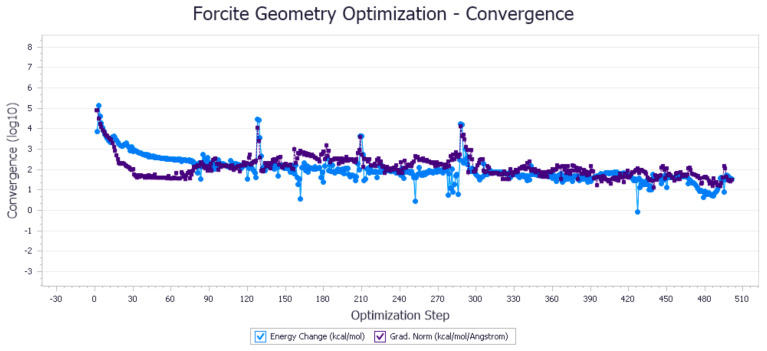
Demulsifier8 # oil–water interface geometric optimization steps.

**Figure 12 molecules-27-01799-f012:**
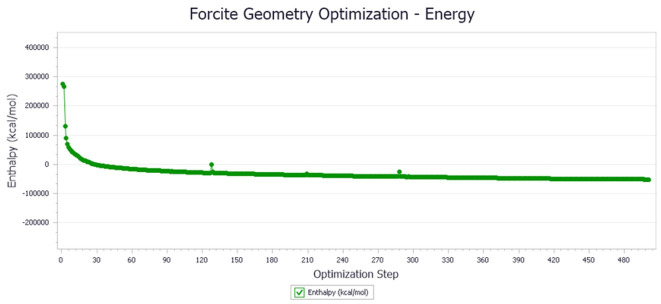
Demulsifier8 # oil–water interface geometry optimization energy trend.

**Figure 13 molecules-27-01799-f013:**
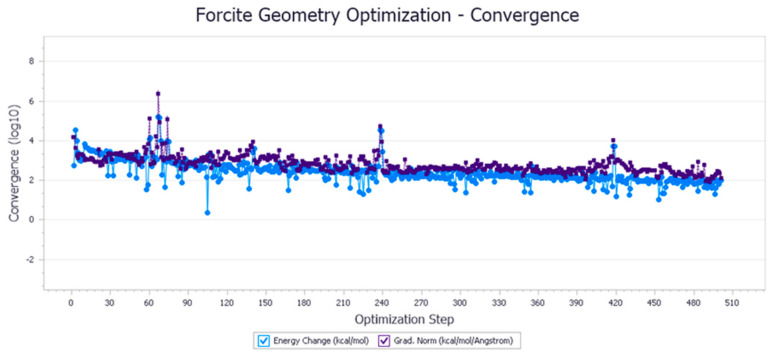
Demulsifier10# oil–water interface geometric optimization steps.

**Figure 14 molecules-27-01799-f014:**
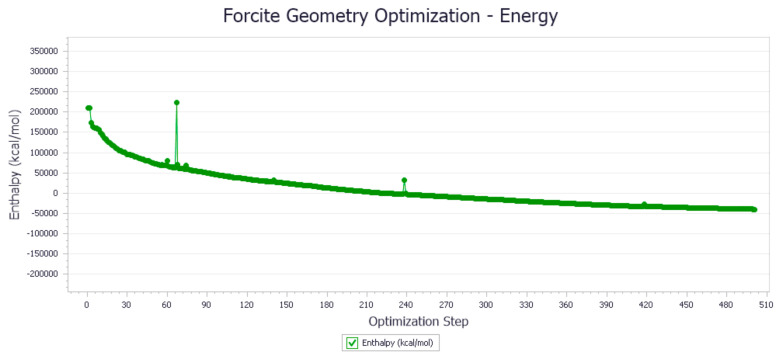
Demulsifier 10 # oil–water interface geometry optimization energy trend.

**Figure 15 molecules-27-01799-f015:**
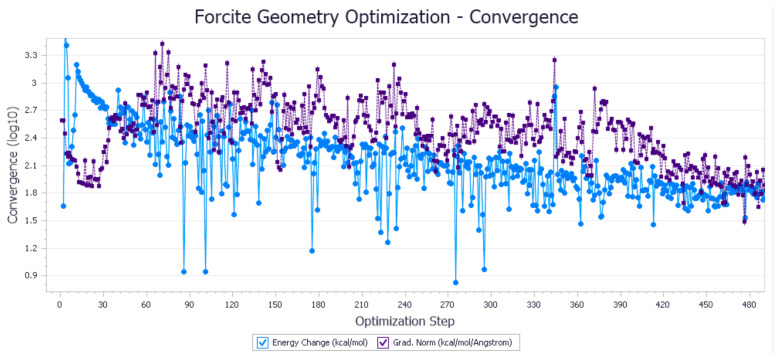
Demulsifier12# oil–water interface geometric optimization steps.

**Figure 16 molecules-27-01799-f016:**
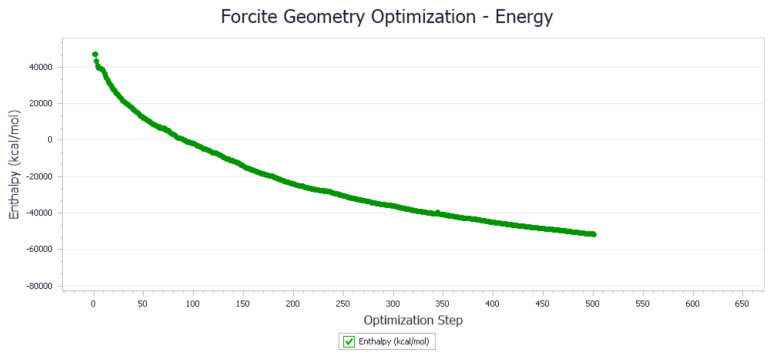
Demulsifier 12 # oil–water interface geometry optimization energy trend.

**Figure 17 molecules-27-01799-f017:**
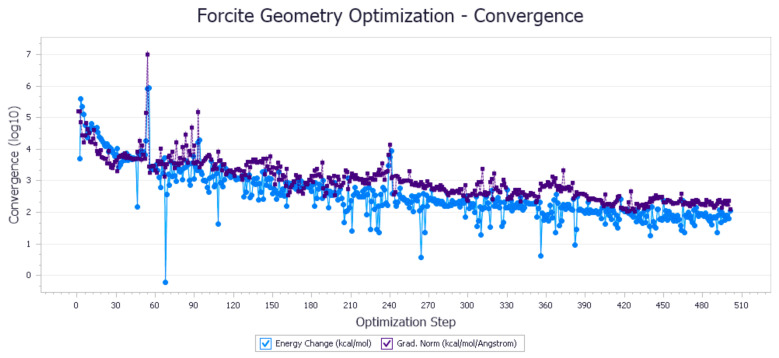
Demulsifier14# oil–water interface geometric optimization steps.

**Figure 18 molecules-27-01799-f018:**
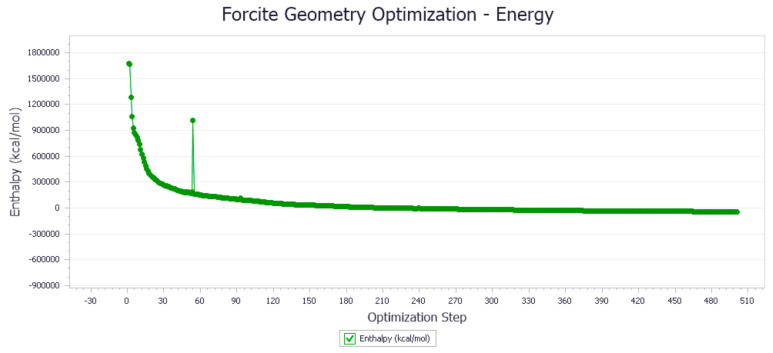
Demulsifier 14 # oil–water interface geometry optimization energy trend.

**Figure 19 molecules-27-01799-f019:**
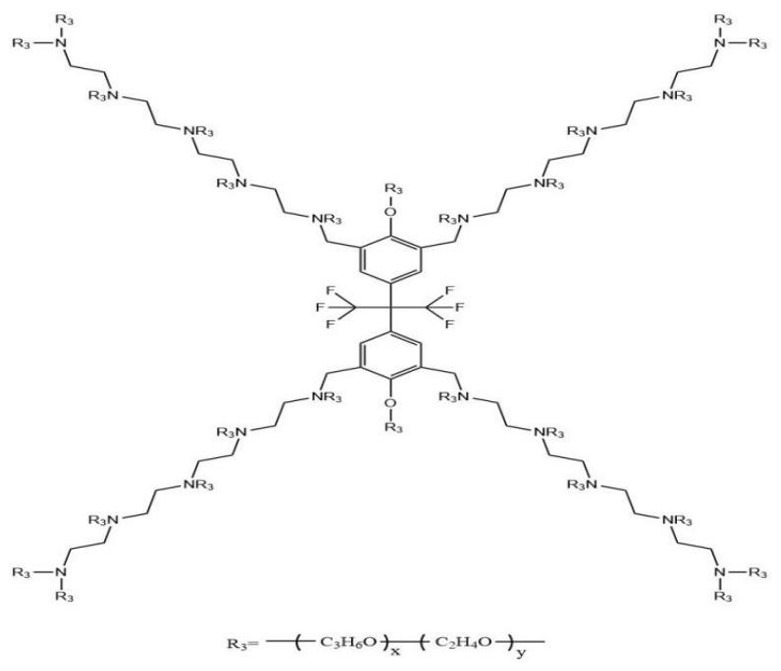
Molecular structure of fluorinated demulsifier.

**Figure 20 molecules-27-01799-f020:**
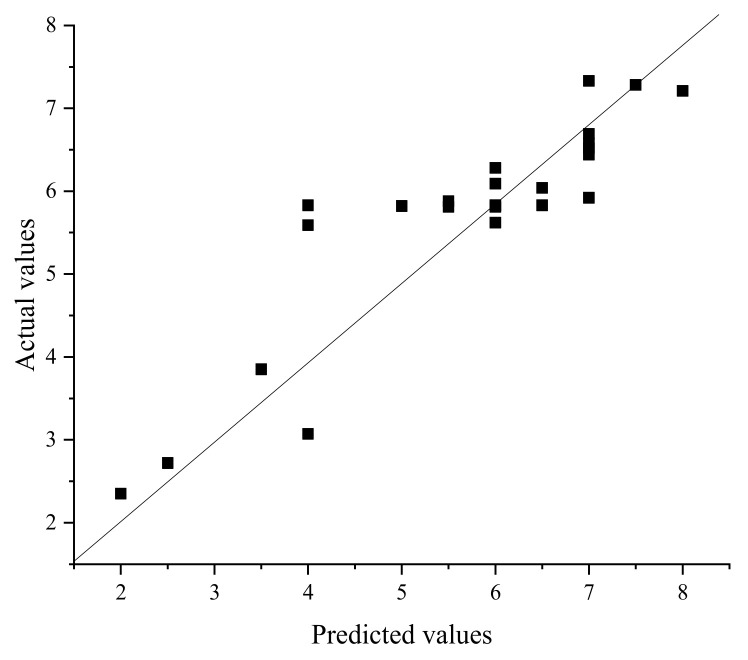
Comparison of predicted and actual values by NAA (r^2^ = 0.802).

**Figure 21 molecules-27-01799-f021:**
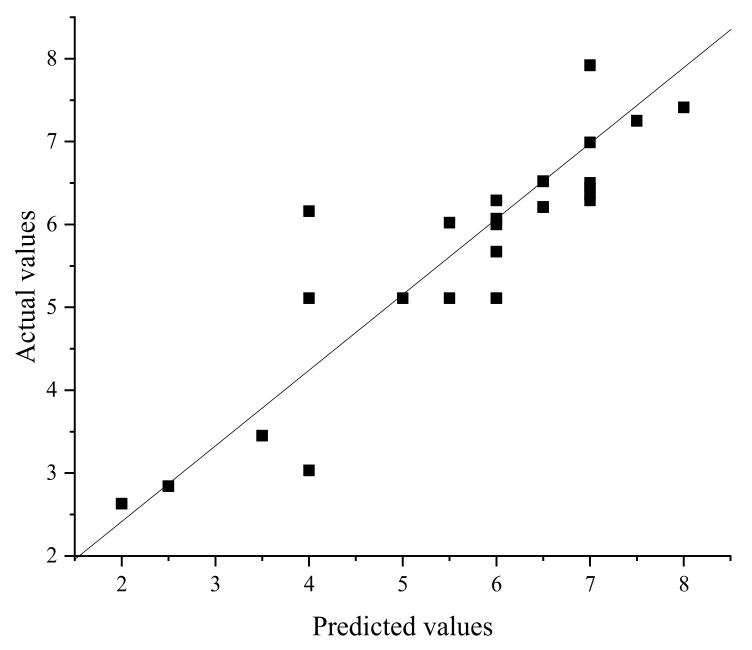
Comparison of predicted and actual values by GFA (r^2^ = 0.861).

**Figure 22 molecules-27-01799-f022:**
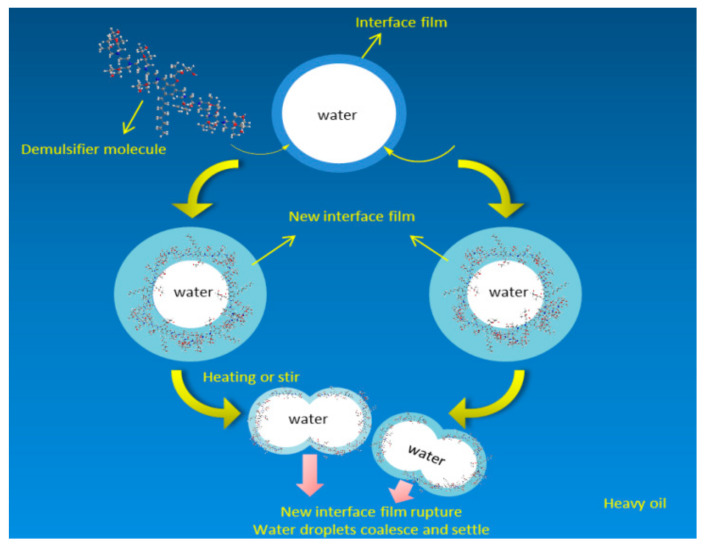
Demulsification mechanism of demulsifier.

**Table 1 molecules-27-01799-t001:** Basic physical properties of crude oil produced from a block from Liaohe Oilfield.

Density kg·m^−3^	Dynamic Viscosity (50 °C) mPa·s	Gum %	Asphaltene %	Acid Value mgKOH·g^−1^	Pour Point °C	Moisture Content %	Saturate %	Aromatic %	Resin %
943.0	180.2	24.93	12.75	2.45	17.3	17.02	46.63	15.69	32.83

**Table 2 molecules-27-01799-t002:** Interfacial generation energy of demulsifier.

Number	Determine of x,y	Water Removal/mL	Etotal/ (Kcal/mol)	IFE/ (Kcal/mol)
1#	SDJ6920: x = 31, y = 20	3.5	−47,819.55	−209.31
2#	SDJ6927: x = 31, y = 15	4	−47,869.13	−214.28
3#	SDJ6930: x = 31, y = 13	2.5	−47,002.77	−127.64
4#	SDJ6937: x = 31, y = 11	2	−46,933.29	−120.69
5#	SDJ9920: x = 44, y = 29	7	−51,139.1	−541.27
6#	SDJ9927: x = 44, y = 21	8	−52,131.12	−640.48
7#	SDJ9930: x = 44, y = 19	7.5	−51,943.12	−621.68
8#	SDJ9937: x = 44, y = 16	7	−51,611.02	−588.47
9#	SDJ15920: x = 71, y = 47	7	−51,002.13	−527.58
10#	SDJ15927: x = 71, y = 35	4	−47,133.09	−140.67
11#	SDJ15930: x = 71, y = 31	6	−50,169.33	−444.30
12#	SDJ15937: x = 71, y = 25	6	−50,129.32	−440.30
13#	SDJ19920: x = 89, y = 59	6	−50,196.02	−447.00
14#	SDJ19927: x = 89, y = 43	7	−51,584.18	−585.78
15#	SDJ19930: x = 89, y = 39	7	−50,113.22	−438.69
16#	SDJ19937: x = 89, y = 32	6.5	−51,008.12	−528.18
17#	SDJ29920: x = 134, y = 88	4	−47,003.72	−127.74
18#	SDJ29927: x = 134, y = 65	5.5	−48,121.12	−239.48
19#	SDJ29930: x = 134, y = 59	7	−51,341.91	−561.55
20#	SDJ29937: x = 134, y = 48	6.5	−51,620.12	−589.38
21#	SDJ39920: x = 178, y = 118	5.5	−49,723.18	−399.68
22#	SDJ39927: x = 179, y = 87	6	−50,339.33	−461.30
23#	SDJ39930: x = 179, y = 79	5	−48,013.13	−228.68
24#	SDJ39937: x = 17, y = 64	6	−50,632.44	−490.61

**Table 3 molecules-27-01799-t003:** Cloud point and HLB value off fluorinated polyether demulsifiers.

Number	Determination of x, y Values	Cloud Point/°C	HLB Value
1#	SDJ6920: x = 31, y = 20	53.2	9.23
2#	SDJ6927: x = 31, y = 15	52.5	9.17
3#	SDJ6930: x = 31, y = 13	52.1	9.13
4#	SDJ6937: x = 31, y = 11	51.8	9.10
5#	SDJ9920: x = 44, y = 29	47.8	8.70
6#	SDJ9927: x = 44, y = 21	47.3	8.66
7#	SDJ9930: x = 44, y = 19	46.9	8.62
8#	SDJ9937: x = 44, y = 16	46.5	8.58
9#	SDJ15920: x = 71, y = 47	41.3	8.07
10#	SDJ15927: x = 71, y = 35	40.7	8.01
11#	SDJ15930: x = 71, y = 31	40.4	7.98
12#	SDJ15937: x = 71, y = 25	40.0	7.94
13#	SDJ19920: x = 89, y = 59	36.2	7.57
14#	SDJ19927: x = 89, y = 43	35.4	7.49
15#	SDJ19930: x = 89, y = 39	35.0	7.45
16#	SDJ19937: x = 89, y = 32	34.7	7.42
17#	SDJ29920: x = 134, y = 88	31.2	7.08
18#	SDJ29927: x = 134, y = 65	30.8	7.04
19#	SDJ29930: x = 134, y = 59	30.3	6.99
20#	SDJ29937: x = 134, y = 48	30.1	6.97
21#	SDJ39920: x = 179, y = 118	28.4	6.80
22#	SDJ39927: x = 179, y = 87	27.9	6.75
23#	SDJ39930: x = 179, y = 79	27.5	6.72
24#	SDJ39937: x = 179, y = 64	27.0	6.67

**Table 4 molecules-27-01799-t004:** The actual water removal amount of demulsifiers (120 min, 100 ppm).

Number	Determination of x, y Values	Water Removal Amount/mL	Demulsification Rate/%
1#	SDJ6920: x = 31, y = 20	3.5	41.13
2#	SDJ6927: x = 31, y = 15	4	47.00
3#	SDJ6930: x = 31, y = 13	2.5	29.38
4#	SDJ6937: x = 31, y = 11	2	23.50
5#	SDJ9920: x = 44, y = 29	7	82.26
6#	SDJ9927: x = 44, y = 21	8	94.01
7#	SDJ9930: x = 44, y = 19	7.5	88.13
8#	SDJ9937: x = 44, y = 16	7	82.26
9#	SDJ15920: x = 71, y = 47	7	82.26
10#	SDJ15927: x = 71, y = 35	4	47.00
11#	SDJ15930: x = 71, y = 31	6	70.51
12#	SDJ15937: x = 71, y = 25	6	70.51
13#	SDJ19920: x = 89, y = 59	6	70.51
14#	SDJ19927: x = 89, y = 43	7	82.26
15#	SDJ19930: x = 89, y = 39	7	82.26
16#	SDJ19937: x = 89, y = 32	6.5	76.38
17#	SDJ29920: x = 134, y = 88	4	47.00
18#	SDJ29927: x = 134, y = 65	5.5	64.63
19#	SDJ29930: x = 134, y = 59	7	82.26
20#	SDJ29937: x = 134, y = 48	6.5	76.38
21#	SDJ39920: x = 179, y = 118	5.5	64.63
22#	SDJ39927: x = 179, y = 87	6	70.51
23#	SDJ39930: x = 179, y = 79	5	58.75
24#	SDJ39937: x = 179, y = 64	6	70.51
Black	No demulsifier	0.3	3.53

**Table 5 molecules-27-01799-t005:** Comparative analysis table between predicted and actual values.

Number	Actual Values	NNA Prediction	Differential Value	GFA Prediction	Differential Value
1#	3.5	3.85	−0.35	3.45	0.04
2#	4	3.07	0.93	3.03	0.97
3#	2.5	2.72	−0.22	2.84	−0.34
4#	2	2.35	−0.35	2.630	−0.630
5#	7	6.69	0.31	7.92	−0.92
6#	8	7.21	0.79	7.41	0.59
7#	7.5	7.28	0.22	7.25	0.25
8#	7	7.33	−0.33	6.99	0.01
9#	7	6.52	0.48	6.36	0.64
10#	4	5.59	−1.59	6.16	−2.16
11#	6	5.62	0.38	6.00	−0.00
12#	6	6.09	−0.09	5.67	0.33
13#	6	6.28	−0.28	6.29	−0.29
14#	7	6.58	0.42	6.50	0.50
15#	7	6.44	0.56	6.44	0.56
16#	6.5	5.83	0.67	6.21	0.29
17#	4	5.83	−1.83	5.11	−1.11
18#	5.5	5.88	−0.38	6.02	−0.52
19#	7	5.92	1.07	6.29	0.71
20#	6.5	6.04	0.46	6.52	−0.020
21#	5.5	5.81	−0.31	5.11	0.39
22#	6	5.81	0.18	5.11	0.89
23#	5	5.82	−0.82	5.11	−0.11
24#	6	5.83	0.17	6.070	−0.07

## Data Availability

All data, models, and code generated or used during the study appear in the submitted article.

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
