# Peer review of "Screening and Demulsification Mechanism of Fluorinated Demulsifier Based on Molecular Dynamics Simulation"

_molecules, 2022, doi:10.3390/molecules27061799_

Round 1

Reviewer 1 Report

The paper" Screening and demulsification mechanism of Fluorinated demulsifiers based on molecular dynamics simulationrequires some of the following modifications:

Please,

  1. Modify the title to Screening and demulsification mechanism of fluorinated demulsifiers based on molecular dynamics simulation. What is the purpose of the second demulsifier mentioned in the title? P1, line 3.
  2. Saturate, aromatic, and resin percentage of Liaohe crude oil should be given in Table 1.
  3. In P1, line 22; 44 and 21 change to 21 (EO) and 44 (PO).
  4. In P1, line 28; R2 change to R2 (superscript), revised in all manuscript.
  5. In this research, crude oil emulsion refers to the type of water-in-oil (w/o) emulsion in which water is the dispersed phase and oil is the continuous phase, not as mentioned (oil in water (o/w) emulsion).
  6. The authors mentioned “Materials” in the experimental section, where are the methods of synthesis of fluorinated polyether demulsifiers, chemical structure confirmation, and preparation scheme?
  7. According to the molecular structure of fluorinated demulsifiers, x represents the number of PO and y represents the number of EO; so, X and y should be corrected in the manuscript.
  8. Calculate the Hydrophile Lipophile Balance (HLB) of fluorinated polyether demulsifiers.
  9. The assessment of the water content in emulsion and the demulsification efficiency of crude oil emulsion in the presence of various demulsifiers should be given. Demulsification efficiency is calculated as follows:

                Efficiency % = (Vo - Vd) / Vo x 100

Where, Vo is the volume of water (water content) in the crude oil emulsion and Vd is the volume of water remaining in the oil phase after demulsifier addition.

  1. Revise the format of the research paper, for example: the x and y axes are not clear and there are some overlaps between the figures and the titles of the figures (e.g. see fig. 5 and the rest).
  2. In P14, lines 331-339; the 2 paragraphs have the same meaning. Please, revise the entire text.
  3. Revise the English language and syntax mistakes along with the manuscript.          

Author Response

Dear Editors and Reviewers:

Thank you for your letter and for the reviewers’ comments concerning our manuscript entitled “Screening and demulsification mechanism of Fluorinated Demulsifier demulsifier based on molecular dynamics simulation” (Manuscript ID: molecules-1558768). Those comments are all valuable and very helpful for revising and improving our paper, as well as having important guiding significance for our research.

According to your comments, we have revised our manuscript carefully, as follows:

Reviewer #1:

Recommendation: The paper" Screening and demulsification mechanism of Fluorinated demulsifiers based on molecular dynamics simulation” requires some of the following modifications.

Comments:

Comment 1: Modify the title to Screening and demulsification mechanism of fluorinated demulsifiers based on molecular dynamics simulation. What is the purpose of the second demulsifier mentioned in the title? P1, line 3.

Response: Thank you for your careful reading of our manuscript. Based on your comments, we have changed the title to “Screening and demulsification mechanism of fluorinated demulsifiers based on molecular dynamics simulation”. The fluorinated demulsifier in the title is not the second demulsifier, it is the general name of the demulsifier used in the simulation study.

Comment 2: Saturate, aromatic, and resin percentage of Liaohe crude oil should be given in Table 1.

Response: Thank you for your careful reading of our manuscript. Based on your comments, the saturation rate, aroma rate and resin rate of Liaohe crude oil are 44.63%, 15.69% and 32.83% respectively in Table 1.

Comment 3: In P1, line 22; 44 and 21 change to 21 (EO) and 44 (PO).

Response: Thank you for your careful reading of our manuscript.Based on your comments, we have changed “44 and 21” change to “21 (EO) and 44 (PO)”.

Comment 4: In P1, line 28; R2 change to R2 (superscript), revised in all manuscript.

Response: Thank you for your careful reading of our manuscript.Based on your comments,we have changed “R2” change to “R2 (superscript)” in all manuscript.

Comment 5: In this research, crude oil emulsion refers to the type of water-in-oil (w/o) emulsion in which water is the dispersed phase and oil is the continuous phase, not as mentioned (oil in water (o/w) emulsion).

Response: Thank you for your careful reading of our manuscript. We are sorry for such a mistake, based on your comments, we changed oil droplets aggregation to water droplets aggregation in the article.

Comment 6:The authors mentioned “Materials” in the experimental section, where are the methods of synthesis of fluorinated polyether demulsifiers, chemical structure confirmation, and preparation scheme?

Response: Thank you for your careful reading of our manuscript. Fluorinated demulsifiers are synthesized using trifluoromethyl phenol, formaldehyde and other raw materials as initiator, and then synthesized by polymerization reaction with propylene oxide and ethylene oxide. The detailed research process was published on ACS OMEGA with reference as Wei, L.; Zhang, L.; Chao, M.;  Jia, X. &  Shi, L. Synthesis and study of a new type of nonanionic demulsifier for chemical flooding emulsion demulsification. ACS Omega. 2021, 6(39), 25518−25528.

Comment 7: According to the molecular structure of fluorinated demulsifiers, x represents the number of PO and y represents the number of EO; so, X and y should be corrected in the manuscript.

Response: Thank you for your careful reading of our manuscript. Based on your comment, we have corrected that x represents the number of PO and y represents the number of EO.

Comment 8: Calculate the Hydrophile Lipophile Balance (HLB) of fluorinated polyether demulsifiers.

Response: Thank you for your careful reading of our manuscript. Based on your comment, we have added experiments on HLB values of fluorinated polyether demulsifiers, and the results are behind 3.1.

The cloud point of fluorinated polyether demulsifier was determined by Cintra 10e UV-Vis spectrometer (GBC Scientific Instruments Company, Australia). The HLB (hydrophilic-lipophilic balance) value of demulsifier was calculated according to the cloud point of surfactant and the empirical formula of HLB to obtain the corresponding HLB value. The empirical formula is as follows :

HLB=0.0980X+4.02

X is the cloud point value of 1wt% fluorinated polyether demulsifier.

The calculation results of HLB value are shown in the Table 3.

Table 3. Cloud point and HLB value off luorinated polyether demulsifiers

Number

Determine of x,y

Cloud point/℃

HLB value

1#

SDJ6920:x=31,y=20

53.2

9.23

2#

SDJ6927:x=31,y=15

52.5

9.17

3#

SDJ6930:x=31,y=13

52.1

9.13

4#

SDJ6937:x=31,y=11

51.8

9.10

5#

SDJ9920:x=44,y=29

47.8

8.70

6#

SDJ9927:x=44,y=21

47.3

8.66

7#

SDJ9930:x=44,y=19

46.9

8.62

8#

SDJ9937:x=44,y=16

46.5

8.58

9#

SDJ15920:x=71,y=47

41.3

8.07

10#

SDJ15927:x=71,y=35

40.7

8.01

11#

SDJ15930:x=71,y=31

40.4

7.98

12#

SDJ15937:x=71,y=25

40.0

7.94

13#

SDJ19920:x=89,y=59

36.2

7.57

14#

SDJ19927:x=89,y=43

35.4

7.49

15#

SDJ19930:x=89,y=39

35.0

7.45

16#

SDJ19937:x=89,y=32

34.7

7.42

17#

SDJ29920:x=134,y=88

31.2

7.08

18#

SDJ29927:x=134,y=65

30.8

7.04

19#

SDJ29930:x=134,y=59

30.3

6.99

20#

SDJ29937:x=134,y=48

30.1

6.97

21#

SDJ39920:x=179,y=118

28.4

6.80

22#

SDJ39927:x=179,y=87

27.9

6.75

23#

SDJ39930:x=179,y=79

27.5

6.72

24#

SDJ39937:x=179,y=64

27.0

6.67

Comment 9: The assessment of the water content in emulsion and the demulsification efficiency of crude oil emulsion in the presence of various demulsifiers should be given. Demulsification efficiency is calculated as follows:

Efficiency % = (VO - Vd) / VO x 100

Where, VO is the volume of water (water content) in the crude oil emulsion and Vd is the volume of water remaining in the oil phase after demulsifier addition.

Response: Thank you for your careful reading of our manuscript. Based on your comments, we have added the formula of demulsification efficiency and provided the water content of crude oil emulsion is 17.02 %.

Comment 10: Revise the format of the research paper, for example: the x and y axes are not clear and there are some overlaps between the figures and the titles of the figures (e.g. see fig. 5 and the rest).

Response: Thank you for your careful reading of our manuscript. We have revised Figure 5 and other pictures according to your requirements.

Comment 11: In P14, lines 331-339; the 2 paragraphs have the same meaning. Please, revise the entire text.

Response: Thank you for your careful reading of our manuscript. Based on your comment, we have changed “IFE value refers to the increase and decrease of the system energy. From Table 2, IFE is negative refers to the decrease of the energy of the whole system. Therefore, after the absolute value analysis of IFE, 6 # demulsifier has the best effect.” to “Table 2 shows that IFE value is negative, indicating that the energy of the whole system decreases. Therefore, after analyzing the absolute value of IFE, 6# demulsifier has the best effect.”

Comment 12: Revise the English language and syntax mistakes along with the manuscript.

Response: Thank you for your careful reading of our manuscript. Based on your comment, we have revised the English language and syntax mistakes.

  1. “The results showed that with the increase of demulsifier concen tration, the kinetic parameters n and t* obtained by characterizing the molecular diffusion decreased”was changed to “The results showed that the kinetic parameters n and t* for molecular diffusion rate decrease with the increase demulsifier concentration”.
  2. “Machine model algorithm can find new rules and development trends from multiple dimensions and a large number of data texts, and integrate and predict them.” was changed to “Machine model algorithm can predict and integrate new rules and development trends from a large number of data texts in multiple dimensions”.
  3. “When the surfactant is dispersed in water, it exists in water as a molecule, and the hydrophobic end of the surfactant is arranged to form a glacier-like structure”was changed to “Surfactants are dispersed in water in a molecular state, and their hydrophobic ends are arranged at the water interface to form a glacier structure, which reduces the entropy of the system. ”
  4. “When the hydrophobic part of the surfactant leaves the water interface, the hydrophobic end of the molecule is associated, resulting in the destruction of the iceberg structure in water and the dissociation of water molecules from the bondage…” was changed to “However, when the hydrophobic end of the surfactant leaves the water interface, the surfactant molecules associate and the glacier structure is destroyed, then the water molecules are separated from the bondage, thereby increasing the entropy of the system.”
  5. “The formation process of micelles is an entropy-driven process, and the chaos of the system increases. The total free energy of formation is negative, which is a spontaneous process." was changed to “The formation process of micelles is a spontaneous entropy-driven process. In this process, the chaos of the system increases, and the total formation energy becomes negative.”
  6. “Substances that ensure oil-water phase dispersion and do not interfere with each other are called oil-water interfacial films.” was changed to “The interface between oil and water is called oil-water interface film”.
  7. “Substances that ensure oil-water phase dispersion and do not interfere with each other are called oil-water interfacial films.” was changed to “The interface between oil and water is called oil-water interface film”.
  8. “Surfactant molecules quickly enter the oil-water emulsion interface membrane, replace the emulsifier molecules” was changed to “Surfactants are added to the emulsion, because of its higher interfacial activity, they replace the natural emulsifier molecules such as asphaltene and colloid adsorbed on the oil-water interfacial film.”
  9. “Under the action of heating or shaking, the macromolecules in the emulsion do irregular Brownian motion and collide with other macromolecules, resulting in the rupture of interfacial film.” was changed to “Under the action of heating or shaking, the Brownian motion of macromolecules in the emulsion is intensified, and the number of collisions between macromolecules is increased. Therefore, the unstable interfacial film formed by demulsifier molecules is broken.”
  10. “ the demulsification mechanism of interfacial film was broken by the collision of fluorinated polyether demulsifier. It was found that under the action of heating or shaking, the macromolecules in the emulsion do irregular Brownian motion and collide with other macromolecules, resulting in the rupture of interfacial film. The water in the internal phase breaks through the interfacial film and enters the external phase to aggregate, so as to achieve the purpose of oil-water separation.” was changed to “the demulsification mechanism of fluorinated polyether demulsifier is that the demulsifier molecules with high interfacial activity will replace the natural emulsifier on the oil-water interfacial film, and form a new unstable interfacial film. Under the action of heating or shaking, the interfacial film collides with other macromolecules, and the interfacial film breaks and the water droplets gather to complete the oil-water separation. ”

Reviewer 2 Report

Comments and Suggestions for Authors

The present article demonstrates the screening of 24 kinds of demulsifiers using the interface generation energy (IFE) module in the molecular dynamics simulation software Materials Studio, and the neural network analysis (NNA) and genetic function approximation (GFA) to predict the best demulsifiers for oil sample from a block in Liaohe Oilfield. There are certain minor comments as given below which need to be addressed before the paper can be recommended for acceptance.

English editing is needed

For example, replace the word demulsification in line 39 to demulsify and the same in line 58

Check the sentences

‘’The results showed that with the increase of demulsifier concen tration, the kinetic parameters n and t* obtained by characterizing the molecular diffusion decreased’’                  line 88

‘’Machine model algorithm can find new rules and development trends from multiple dimensions and a large number of data texts, and integrate and predict them’’             Line 93

In the section 2.5.

Did you consider a blank sample without demulsifier addition? If yes please state that in the section.

In Table 2. Could you add a column of demulsification percentage?

Author Response

Dear Editors and Reviewers:

Thank you for your letter and for the reviewers’ comments concerning our manuscript entitled “Screening and demulsification mechanism of Fluorinated Demulsifier demulsifier based on molecular dynamics simulation” (Manuscript ID: molecules-1558768). Those comments are all valuable and very helpful for revising and improving our paper, as well as having important guiding significance for our research.

According to your comments, we have revised our manuscript carefully, as follows:

Reviewer #2:

Recommendation: The present article demonstrates the screening of 24 kinds of demulsifiers using the interface generation energy (IFE) module in the molecular dynamics simulation software Materials Studio, and the neural network analysis (NNA) and genetic function approximation (GFA) to predict the best demulsifiers for oil sample from a block in Liaohe Oilfield. There are certain minor comments as given below which need to be addressed before the paper can be recommended for acceptance.

Comments:

Comment 1: English editing is needed. For example, replace the word demulsification in line 39 to demulsify and the same in line 58.

Response: Thank you for your careful reading of our manuscript. Based on your comments, we have changed the word “demulsification” to “demulsify” in line 39 and line 58. 

Comment 2: Check the sentences

“The results showed that with the increase of demulsifier concen tration, the kinetic parameters n and t* obtained by characterizing the molecular diffusion decreased’’  line 88

“Machine model algorithm can find new rules and development trends from multiple dimensions and a large number of data texts, and integrate and predict them’’  Line 93

Response: Thank you for your careful reading of our manuscript. Based on your comments, we have changed “The results showed that with the increase of demulsifier concen tration, the kinetic parameters n and t* obtained by characterizing the molecular diffusion decreased” to “The results showed that the kinetic parameters n and t* for molecular diffusion rate decrease with the increase demulsifier concentration”.

“Machine model algorithm can find new rules and development trends from multiple dimensions and a large number of data texts, and integrate and predict them.” was changed to “Machine model algorithm can predict and integrate new rules and development trends from a large number of data texts in multiple dimensions”.

Comment 3: In the section 2.5.Did you consider a blank sample without demulsifier addition? If yes please state that in the section.

Response: Thank you for your careful reading of our manuscript.We took into account the establishment of the blank sample without demulsifier addition and obtained the dehydration Vb. After adding demulsifier, the dehydration was V, and then the dehydration of demulsifier Vd = V − Vb. Therefore, the subsequent dehydration of the paper is Vd. We also added a row of dehydration without demulsifier in Table 3.

Comment 4: In Table 2. Could you add a column of demulsification percentage?

Response: Thank you for your careful reading of our manuscript.Based on your comments,we have added a column of demulsification percentage.

Table 4. The actual water removal amount of demulsifiers (120min, 100ppm).

Number

Determine of x,y

water removal amount/ml

Demulsification rate/%

1#

SDJ6920:x=31,y=20

3.5

41.13

2#

SDJ6927:x=31,y=15

4

47.00

3#

SDJ6930:x=31,y=13

2.5

29.38

4#

SDJ6937:x=31,y=11

2

23.50

5#

SDJ9920:x=44,y=29

7

82.26

6#

SDJ9927:x=44,y=21

8

94.01

7#

SDJ9930:x=44,y=19

7.5

88.13

8#

SDJ9937:x=44,y=16

7

82.26

9#

SDJ15920:x=71,y=47

7

82.26

10#

SDJ15927:x=71,y=35

4

47.00

11#

SDJ15930:x=71,y=31

6

70.51

12#

SDJ15937:x=71,y=25

6

70.51

13#

SDJ19920:x=89,y=59

6

70.51

14#

SDJ19927:x=89,y=43

7

82.26

15#

SDJ19930:x=89,y=39

7

82.26

16#

SDJ19937:x=89,y=32

6.5

76.38

17#

SDJ29920:x=134,y=88

4

47.00

18#

SDJ29927:x=134,y=65

5.5

64.63

19#

SDJ29930:x=134,y=59

7

82.26

20#

SDJ29937:x=134,y=48

6.5

76.38

21#

SDJ39920:x=179,y=118

5.5

64.63

22#

SDJ39927:x=179,y=87

6

70.51

23#

SDJ39930:x=179,y=79

5

58.75

24#

SDJ39937:x=179,y=64

6

70.51

Black

No demulsifier

0.3

3.53

Reviewer 3 Report

The manuscript presents the data from computational simulation screening about the efficiency of a number of demulsifiers on the performance of Liaohe crude oil emulsion. Based on this the substance with the best demulsification effect is defined.

The obtained results are of interest to the readers of MDPI Molecules, but there are issues which have to be resolved before accepting the manuscript:

  1. The choice of the particular computational procedures should be better substantiated and presented in more details as related to the specific properties of Liaohe crude oil emulsion.
  2. The experiments on demulsification and water removal of the demulsifier, as presented in section 2.5 (lines 257-268), are vaguely described; the experimental conditions are not clearly related to the overall study aim.
  3. There are ambiguous statements and unclear/wrong terminology, e.g.:

(a) lines 308-317:

- “When the surfactant is dispersed in water, it exists in water as a molecule, and the hydrophobic end of the surfactant is arranged to form a glacier-like structure…”;

- “When the hydrophobic part of the surfactant leaves the water interface, the hydrophobic end of the molecule is associated, resulting in the destruction of the iceberg structure in water and the dissociation of water molecules from the bondage…”;

- “The formation process of micelles is an entropy-driven process, and the chaos of the system increases. The total free energy of formation is negative, which is a spontaneous process."

(b) lines 363-374:

- “Substances that ensure oil-water phase dispersion and do not interfere with each other are called oil-water interfacial films.”;

- “….one substance stably adheres to the surface of another substance, forming an interface layer which has a protective effect on solute”;

- “The interface layer has low surface tension and interface free energy, and its relative concentration is higher than that of solute concentration.”;

- “The demulsification mechanism of fluorinated polyether demulsifier in this study is mainly the mechanism of breaking the interface film. With the large-scale use of polymer demulsifiers, the mechanism of breaking interfacial film is increasingly recognized by a large number of researchers.”

(c) line 383-384

- “Surfactant molecules quickly enter the oil-water emulsion interface membrane, replace the emulsifier molecules”

(d) lines 389-391

- “Under the action of heating or shaking, the macromolecules in the emulsion do irregular Brownian motion and collide with other macromolecules, resulting in the rupture of interfacial film.”

(e) lines 411-416

- “… the demulsification mechanism of interfacial film was broken by the collision of fluorinated polyether demulsifier. It was found that under the action of heating or shaking, the macromolecules in the emulsion do irregular Brownian motion and collide with other macromolecules, resulting in the rupture of interfacial film. The water in the internal phase breaks through the interfacial film and enters the external phase to aggregate, so as to achieve the purpose of oil-water separation.”

Author Response

Dear Editors and Reviewers:

Thank you for your letter and for the reviewers’ comments concerning our manuscript entitled “Screening and demulsification mechanism of Fluorinated Demulsifier demulsifier based on molecular dynamics simulation” (Manuscript ID: molecules-1558768). Those comments are all valuable and very helpful for revising and improving our paper, as well as having important guiding significance for our research.

According to your comments, we have revised our manuscript carefully, as follows:

Reviewer #3:

The manuscript presents the data from computational simulation screening about the efficiency of a number of demulsifiers on the performance of Liaohe crude oil emulsion. Based on this the substance with the best demulsification effect is defined.The obtained results are of interest to the readers of MDPI Molecules, but there are issues which have to be resolved before accepting the manuscript.

Comment 1:The choice of the particular computational procedures should be better substantiated and presented in more details as related to the specific properties of Liaohe crude oil emulsion.

Response: Thank you for your careful reading of our manuscript. Based on your comments, we added  the saturation rate, aroma rate and resin rate of Liaohe crude oil oil in Table 1, which were 44.63 %, 15.69 % and 32.83 %, respectively.

Comment 2: The experiments on demulsification and water removal of the demulsifier, as presented in section 2.5 (lines 257-268), are vaguely described; the experimental conditions are not clearly related to the overall study aim.

Response: Thank you for your careful reading of our manuscript. Based on your comments, we give a more detailed description of demulsification experiments, as follows:

Put crude oil emulsion with water content of 17.02 % into constant temperature water bath heated to 55 °C for 30 min, and then put into a stirring motor for 8min at a speed of 2000r/min. After that, it was put into the stirring machine for 5min. Pour 50ml crude oil emulsion into a calibrated test tube and put it into a water bath heated to 60℃ and kept at a constant temperature for 25min. The height of the water surface should not exceed the height of the crude oil in the test tube. Add demulsifier into the test tube with micropipeter and tighten the cork. Turn the test tube upside down and shake it 3~5 times, loosen the cork and let off air. Recork the bottle and shake the tube 150 times with your hand to fully mix the demulsifier and crude oil emulsion. After the cork is capped, the bottle is placed in a water bath at 60℃ for settling. The volume of dehydration at different times was observed to obtain the dehydration volume V. The blank sample without demulsifier addition was set to obtain the dehydration amount Vb. Therefore, the dehydration amount after adding demulsifier was Vd = V-Vb.

Demulsification efficiency is calculated as follows:

Efficiency (%) = (VO- Vd) / VO x 100

Where,VO is the volume of water (water content) in the crude oil emulsion and Vd is the volume of water remaining in the oil phase after demulsifier addition.

The experimental conditions in demulsification and dehydration were based on the bottle test method, and the demulsification effect was relatively scientific. The demulsification efficiency obtained by experiments can be compared with the demulsification efficiency predicted by the two models, so as to determine the accuracy of the model prediction and screen the optimal demulsifier.

Comment 3: There are ambiguous statements and unclear/wrong terminology, e.g.:

(a) lines 308-317:

- “When the surfactant is dispersed in water, it exists in water as a molecule, and the hydrophobic end of the surfactant is arranged to form a glacier-like structure…”;

- “When the hydrophobic part of the surfactant leaves the water interface, the hydrophobic end of the molecule is associated, resulting in the destruction of the iceberg structure in water and the dissociation of water molecules from the bondage…”;

- “The formation process of micelles is an entropy-driven process, and the chaos of the system increases. The total free energy of formation is negative, which is a spontaneous process."

(b) lines 363-374:

- “Substances that ensure oil-water phase dispersion and do not interfere with each other are called oil-water interfacial films.”;

- “….one substance stably adheres to the surface of another substance, forming an interface layer which has a protective effect on solute”;

- “The interface layer has low surface tension and interface free energy, and its relative concentration is higher than that of solute concentration.”;

- “The demulsification mechanism of fluorinated polyether demulsifier in this study is mainly the mechanism of breaking the interface film. With the large-scale use of polymer demulsifiers, the mechanism of breaking interfacial film is increasingly recognized by a large number of researchers.”

(c) line 383-384

- “Surfactant molecules quickly enter the oil-water emulsion interface membrane, replace the emulsifier molecules”

(d) lines 389-391

- “Under the action of heating or shaking, the macromolecules in the emulsion do irregular Brownian motion and collide with other macromolecules, resulting in the rupture of interfacial film.”

(e) lines 411-416

- “… the demulsification mechanism of interfacial film was broken by the collision of fluorinated polyether demulsifier. It was found that under the action of heating or shaking, the macromolecules in the emulsion do irregular Brownian motion and collide with other macromolecules, resulting in the rupture of interfacial film. The water in the internal phase breaks through the interfacial film and enters the external phase to aggregate, so as to achieve the purpose of oil-water separation.”

Response: Thank you for your careful reading of our manuscript. Based on your comments, we have corrected the above sentences.

(a) lines 308-317:

- “When the surfactant is dispersed in water, it exists in water as a molecule, and the hydrophobic end of the surfactant is arranged to form a glacier-like structure…”; was changed to “Surfactants are dispersed in water in a molecular state, and their hydrophobic ends are arranged at the water interface to form a glacier structure, which reduces the entropy of the system. ”

- “When the hydrophobic part of the surfactant leaves the water interface, the hydrophobic end of the molecule is associated, resulting in the destruction of the iceberg structure in water and the dissociation of water molecules from the bondage…”; was changed to “However, when the hydrophobic end of the surfactant leaves the water interface, the surfactant molecules associate and the glacier structure is destroyed, then the water molecules are separated from the bondage, thereby increasing the entropy of the system.”

- “The formation process of micelles is an entropy-driven process, and the chaos of the system increases. The total free energy of formation is negative, which is a spontaneous process." was changed to “The formation process of micelles is a spontaneous entropy-driven process. In this process, the chaos of the system increases, and the total formation energy becomes negative.”

(b) lines 363-374:

- “Substances that ensure oil-water phase dispersion and do not interfere with each other are called oil-water interfacial films.”; was changed to “The interface between oil and water is called oil-water interface film”

- “….one substance stably adheres to the surface of another substance, forming an interface layer which has a protective effect on solute”; and - “The interface layer has low surface tension and interface free energy, and its relative concentration is higher than that of solute concentration.”; were changed to “Natural emulsifiers such as asphaltene and colloid in crude oil emulsion are stably adsorbed on the surface of water droplets, forming an interfacial film with low surface tension and interfacial free energy.”

(c) line 383-384

- “Surfactant molecules quickly enter the oil-water emulsion interface membrane, replace the emulsifier molecules” was changed to “Surfactants are added to the emulsion, because of its higher interfacial activity, they replace the natural emulsifier molecules such as asphaltene and colloid adsorbed on the oil-water interfacial film.”

(d) lines 389-391

- “Under the action of heating or shaking, the macromolecules in the emulsion do irregular Brownian motion and collide with other macromolecules, resulting in the rupture of interfacial film.” was changed to “Under the action of heating or shaking, the Brownian motion of macromolecules in the emulsion is intensified, and the number of collisions between macromolecules is increased. Therefore, the unstable interfacial film formed by demulsifier molecules is broken.”

(e) lines 411-416

- “… the demulsification mechanism of interfacial film was broken by the collision of fluorinated polyether demulsifier. It was found that under the action of heating or shaking, the macromolecules in the emulsion do irregular Brownian motion and collide with other macromolecules, resulting in the rupture of interfacial film. The water in the internal phase breaks through the interfacial film and enters the external phase to aggregate, so as to achieve the purpose of oil-water separation.” was changed to “the demulsification mechanism of fluorinated polyether demulsifier is that the demulsifier molecules with high interfacial activity will replace the natural emulsifier on the oil-water interfacial film, and form a new unstable interfacial film. Under the action of heating or shaking, the interfacial film collides with other macromolecules, and the interfacial film breaks and the water droplets gather to complete the oil-water separation. ”